# The Influence of the Microbiome on Immunotherapy for Gastroesophageal Cancer

**DOI:** 10.3390/cancers15184426

**Published:** 2023-09-05

**Authors:** Neda Dadgar, Vinay Edlukudige Keshava, Moses S. Raj, Patrick L. Wagner

**Affiliations:** 1Cole Eye Institute, Cleveland Clinic, Cleveland, OH 44106, USA; dadgarn@ccf.org; 2Allegheny Health Network Cancer Institute, Pittsburgh, PA 15224, USA; vinay.edlukudigekeshava@ahn.org (V.E.K.); moses.raj@ahn.org (M.S.R.)

**Keywords:** gastroesophageal cancers, PD1 blockade, immunotherapy, resistance, prognosis, survival, biomarkers, microbiome

## Abstract

**Simple Summary:**

This article delves into the mounting impact of the microbiome on gastroesophageal cancer. Investigating its connection to immunotherapy, this study explores microbial shifts within tumor environments, the gastrointestinal tract, and their consequential effects. Notable associations have surfaced, linking dysbiosis, reduced diversity, treatment resistance, and distinct microbe profiles to improved treatment responses. Comprehending these intricate dynamics holds the key to tailored interventions. The potential of microbiome-based biomarkers is promising for personalized treatment strategies and response prediction. Discussions encompass interventions such as microbiota-based therapeutics and dietary adjustments, which could amplify immunotherapy efficacy by shaping the gut environment. However, realizing the full potential of microbiome-centric approaches hinges on thorough research and substantial clinical trials. This article highlights the evolving role of the microbiome and its dual impact on both gastroesophageal cancer and the efficacy of immunotherapy in reshaping future treatment paradigms.

**Abstract:**

Immunotherapy has shown promise as a treatment option for gastroesophageal cancer, but its effectiveness is limited in many patients due to the immunosuppressive tumor microenvironment (TME) commonly found in gastrointestinal tumors. This paper explores the impact of the microbiome on the TME and immunotherapy outcomes in gastroesophageal cancer. The microbiome, comprising microorganisms within the gastrointestinal tract, as well as within malignant tissue, plays a crucial role in modulating immune responses and tumor development. Dysbiosis and reduced microbial diversity are associated with poor response rates and treatment resistance, while specific microbial profiles correlate with improved outcomes. Understanding the complex interactions between the microbiome, tumor biology, and immunotherapy is crucial for developing targeted interventions. Microbiome-based biomarkers may enable personalized treatment approaches and prediction of patient response. Interventions targeting the microbiome, such as microbiota-based therapeutics and dietary modifications, offer the potential for reshaping the gut microbiota and creating a favorable TME that enhances immunotherapy efficacy. Further research is needed to reveal the underlying mechanisms, and large-scale clinical trials will be required to validate the efficacy of microbiome-targeted interventions.

## 1. Introduction

Although progress has been made in reducing the incidence of esophageal cancer (EC) and gastric cancer (GC), they still contribute significantly to global cancer mortality [1,2]. Reducing exposure to known risk factors, regular screening of high-risk individuals, and early detection can help decrease the incidence of GC and EC, but most patients continue to present with advanced disease, for which conventional treatments offer little chance of a cure [3,4]. Thus, there has been strong interest in adapting novel treatment strategies to EC and GC, with significant progress in the integration of immunotherapy into modern treatment algorithms [5]. As a result, current NCCN guidelines now endorse immunotherapy as an option for several key clinical scenarios in EC and GC, including first-line treatment for metastatic esophageal or gastroesophageal junction (GEJ) adenocarcinoma (in combination with conventional cytotoxic therapy), second-line treatment of unresectable or metastatic esophageal squamous-cell carcinoma, and post-operative treatment for selected high-risk patients with esophageal or GEJ adenocarcinoma following neoadjuvant chemo-radiation and surgery [6]. 

Since only a minority of patients are eligible for or responsive to immunotherapy, there is an intense need to develop new strategies for expanding the pool of patients in whom immunotherapy is expected to provide a survival benefit. One emerging strategy involves harnessing the tumor and host microbiome to promote anti-tumor immunity and immunotherapy, based on several key lines of evidence linking the microbiome with cancer biology and immune response. First, dysbiosis and reduced microbial diversity have been associated with poor response rates and treatment resistance in gastroesophageal cancer patients, while specific microbial profiles have shown correlation with improved outcomes [7,8]. Second, microbiome-based biomarkers have shown promise in predicting patient response to immunotherapy [9]. Third, interventions targeting the microbiome, such as microbiota transfer techniques, prebiotics, and dietary modifications, are gaining increasing traction due to their proven ability to enhance immunotherapy efficacy [10]. Understanding the complex interactions between the microbiome, tumor biology, and immunotherapy is thus essential in order to improve upon the benefits that immunotherapy has brought to patients with EC/GC.

In this review article, we discuss the influence of the tumor and host microbiome in EC/GC pathogenesis and tumor immune microenvironment, existing approved immunotherapy options for EC and GC, and the relationship between microbiome constituents and immunotherapy efficacy. As knowledge in this area evolves, it is hoped that new insights will lead to increasingly effective immunotherapy options for patients with EC and GC.

## 2. Tumor and Host Microbiome on the Pathogenesis of EC and GC

### 2.1. Microbiome in Esophageal Cancer

Esophageal cancer (EC) is a highly aggressive malignancy, comprising two main subtypes, esophageal squamous-cell carcinoma/ESCC and esophageal adenocarcinoma/EAC [11]. EAC is linked to risk factors such as a Western diet, obesity, gastroesophageal reflux disease (GERD), and Barrett’s esophagus (BE), which can lead to EAC through progressive dysplasia [12,13]. By contrast, ESCC is more strongly associated with smoking and alcohol use [14,15]. A growing body of research has documented the correlation of ESCC and EAC incidence with unique microbial signatures from the oral mucosa, esophageal lumen, tumor tissue, or gut (Figure 1). These data provide compelling epidemiologic evidence of an association between the microbiome and EC, and may open new pathways for research into microbiome-based prevention or treatment strategies.

The **oral cavity**’s normal bacterial flora are primarily composed of six major phyla, namely, *Firmicutes, Bacteroidetes, Proteobacteria, Actinobacteria, Spirochaetes,* and *Fusobacteria* [16], with the most frequent species being *Streptococcus* spp. *(S. mitis*, *S. sanguis, S. gordonii*), *Gemella* spp. *(G. sanguinis*, *G. haemolysans), Granulicatella* spp., and *Neisseria* spp. [17]. Recent studies suggest an association between ESCC and increased oral abundance of *Firmicutes*, *Prevotella, Streptococcus*, and *Porphyromonas*, as well as a decrease in *Proteobacteria*, *Neisseriales, Lautropia, Bulleidia, Catonella, Corynebacterium, Moryella, Peptococcus*, and *Cardiobacterium* [18,19,20,21,22]. Similar but distinct changes have been observed in EAC, where *Tannerella forsythia* was associated with higher risk, while depletion of *Neisseria* and *Streptococcus pneumoniae* was associated with lower risk [22,23]. These differences in the oral microbiome could serve as a biomarker for predicting esophageal cancer risk or directly influence cancer risk by mechanisms that remain to be discovered [24].

The **gastric microbiome** may also impact or predict esophageal cancer, and special mention is warranted for *Helciobacter pylori* in any discussion of microbial influences on upper gastrointestinal cancer. A meta-analysis aimed at clarifying the link between *H. pylori* infection and esophageal carcinoma found no significant association with ESCC in the overall population. However, significant associations were found in Eastern subjects, suggesting a decreased risk of ESCC in individuals infected with cagA-positive *H. pylori* strains [25]. Another study aimed to compare the pattern of gastric corpus microbiota in early ESCC and esophageal squamous dysplasia with normal esophagus. *Clostridiales* and *Erysipelotrichales* orders were more abundant among cases. No such difference was observed between esophagitis and healthy controls, suggesting that the composition of gastric corpus mucosal microbiota may differ along the spectrum of dysplasia and cancer in ESCC [26]. In another study, the gastric juice samples of EC patients showed a significant reduction in the abundance of genera *Siphonobacter, Balneola, Nitrosopumilus,* and *Planctomyces* [27].

**The gut microbiome,** consisting predominantly of the phyla *Firmicutes, Bacteriodetes, Proteobacteria*, and *Actinobacteria* [28], may also influence EC pathogenesis. Microbiota can stimulate carcinogenesis through immune pathways, as Münch et al. showed in a mouse model of Barrett’s esophagus. High-fat diet-induced changes in gut microbiota led to proinflammatory cytokines, immune cells, and a pro-tumor immune phenotype, highlighting the critical role of gut microbiota in transmitting dietary influences via inflammatory mechanisms [29]. In ESCC patients, Yang et al. found that the levels of *Bacteroidetes, Fusobacteria*, and *Spirochaetes* in the gut microbiome are decreased relative to controls [30]. Another study showed that the abundance of short-chain fatty acid-producing bacteria is reduced, while the amount of lipopolysaccharide-producing bacteria increases in EC patients [31]. In a similar study, the fecal quantities of *Bacteroides fragilis*, *Escherichia coli*, *Akkermansia muciniphila, Clostridium hathaway*, and *Alistipes finegoldii* distinguished cancer patients versus control subjects [32]. In a study of 783 patients who underwent oncologic esophagectomy, researchers investigated the gut microbiota detected by fecal culture tests and its association with patient outcomes, finding that *Bacillus* species had a better response to preoperative treatment, lower modified Glasgow prognostic score, and improved survival, whereas patients with *P. mirabilis* had higher systemic inflammation scores, increased postoperative pneumonia incidence, and unfavorable survival [33]. The mechanisms accounting for these intriguing differences remain unknown.

The **intratumoral microbiome** present within malignant ESCC, EAC, and BE tissue has uniformly been found to show less diversity relative to normal esophageal tissue [27,34]. EAC was associated with increased abundance of intratumoral *Streptococcus* and *Neisseria*, and the microbiome composition of EAC differed from that of Barrett’s esophagus and GERD, suggesting that local dysbiosis may be associated with neoplastic progression along the Barrett’s sequence [35]. Interestingly, some studies have found that certain bacterial species associated with periodontal disease, such as *Fusobacterium nucleatum* and *Tannerella forsythia*, have been detected in esophageal cancer tissues, suggesting that oral microbiota may contribute to esophageal cancer pathogenesis [36,37,38]. The analysis of microbial composition has been examined as a means of stratifying cancer risk, with studies indicating certain bacteria as strong predictors for the development of ESCC and EAC. For ESCC, bacteria such as *P gingivalis*, *Streptococcus*, *Neisseria*, *Actinomyces*, and *Atopobium* are associated with a higher risk, whereas *Prevotella* oral *taxon 306* and *Aggregatibacter paraphrophilus* suggest a lower risk [39]. EAC exhibited a remarkable decrease in *Streptococcus*, accompanied by an increase in *Prevotella, Veillonella,* and *Leptotrichia* [40]. In the case of EAC, the periodontal pathogen *Tannerella* forsythia and oral species such as *Actinomyces cardiffensis, Selenomonas* oral taxon 134, and *Veillonella* oral *taxon 917* are potential factors for a higher risk, while certain bacteria such as *Lachnoanaerobaculum umeaense, Oribacterium parvum, Solobacterium moorei (Firmicutes), Neisseria sicca, Neisseria flavescens*, and *Haemophilus* oral *taxon 908* (*Proteobacteria*), *Corynebacterium durum*, *Prevotella nanceiensis*, and *S. pneumoniae* are associated with a lower risk. However, it remains unclear whether these microbial changes play a causative role in cancer development or are a consequence of the presence of cancer [23,41].

A few studies focusing on surgical resection specimens have examined the esophageal microbiome and metabolic changes before and after esophagectomy. One such study aimed to characterize the esophageal microbiome of patients with esophageal squamous-cell carcinoma (ESCC) and GEJ cancer, as well as post-esophagectomy patients, versus healthy controls (HC). Microbial diversity was significantly lower in the ESCC, GEJ, and post-ESCC groups than in the HC group. The abundance of *Fusobacteria* was higher, and the abundance of *Actinobacteria* was lower in the ESCC group than in the HC group. Significant differences were found in the abundance of *Bacteroidetes* and *Fusobacteria* between the ESCC and post-ESCC group. Microbial sequencing results from 19 pairs of tissues revealed that *Proteobacteria, Firmicutes, Bacteroidetes, Deinococcus-Thermus*, and *Actinobacteria* were the predominant bacteria in both tumor and adjacent non-tumor tissues. Additionally, the group found that *Streptococcus* had the highest relative proportion in tumor tissues, while *Labrys* was more abundant in adjacent non-tumor tissues at the genus level. It was also observed that the microbial interactions in tumor tissues were less complex than those in adjacent non-tumor tissues, providing valuable insights into the potential relationship between intratumoral microorganisms and esophageal carcinogenesis [42,43].

### 2.2. Microbiome in Gastric Cancer

Gastric cancer (GC) has been identified as a disease caused by a combination of genetic, molecular, and environmental factors, with *H. pylori* infection being the most common factor among them [44,45,46]. Recent discoveries challenge previous assumptions about the uniformity of microhabitats within the stomach, revealing significant variations in pH, mucin distribution, nutrients, ions, and chemical levels between tumor and adjacent tumor-free tissue [47]. In GC patients, five **oral bacteria** were found to accurately distinguish GC from non-atrophic gastritis with good performance. The increase in oral abundance of *H. pylori*, *Shigella* spp., *Lactobacillus* spp., *Atopobium* spp., *Megasphaera* spp., *Streptococcus* spp., *Veillonella* spp., *Prevotella* spp., and *Clostridium* spp. flora in GC, and significant enrichment of *Escherichia coli*, *Prevotella* spp., *Clostridium* spp., and *Bacteroides fragilis* have also been associated with gastric tumorigenesis. The abundance of *Sphingobium yanoikuyae* was significantly reduced in GC, and higher relative abundances of several bacterial genera commonly found in the oral cavity, including *Fusobacterium, Veillonella, Leptotrichia, Haemophilus Campylobacter*, have been observed in gastric cancer patients [48].

These variations impact the diversity and composition of the **gastric microbiome**, which is primarily made up of five phyla, namely, *Proteobacteria*, *Firmicutes*, *Bacteroidetes*, *Actinobacteria*, and *Fusobacteria*. Notably, *H. pylori* has been identified as the most prevalent species among GC tumor samples. However, following surgery, there is a considerable shift in the gastric microbiome’s community composition at the phylum level, with a reduction in *Proteobacteria* and *Actinobacteria* and an increase in *Firmicutes* and *Bacteroidetes* [49]. A study conducted by Park et al. analyzed gastric secretions from patients with gastritis, gastric adenoma, or gastric cancer, finding that microbial diversity decreased continuously along the spectrum of gastric carcinogenesis, and the microbial composition differed significantly by disease status. The composition of the gastric microbiome in patients with gastric cancer was characterized by reduced levels of *Verrucomicrobia* and *Deferribacteres*, whereas *Akkermansia* and *Lachnospiraceae NK4A136* group were significantly more abundant in patients with gastritis. The study also predicted functional pathways related to carcinogenesis using Tax4Fun, suggesting that gastric cancer is associated with microbial dysbiosis and functional changes that could promote carcinogenesis [50].

A study conducted by Liang et al. compared the **fecal microbiota** of GC patients before and after radical distal gastrectomy with that of healthy individuals. The study found that during the perioperative period, the relative abundances of certain genera, such as *Akkermansia*, *Esherichia/Shigella*, *Lactobacillus*, and *Dialister*, showed significant changes. Moreover, GC patients exhibited higher abundances of *Escherichia/Shigella*, *Veillonella*, and *Clostridium XVIII* and lower abundances *of Bacteroides* when compared to healthy controls. Another study showed increased abundance of *H. pylori, Shigella* spp., *Lactobacillus* spp., *Atopobium* spp., *Megasphaera* spp., *Streptococcus* spp., *Veillonella* spp., *Prevotella* spp., and *Clostridium* spp. flora in gut microbia in the patients with gastric cancer, and significant enrichment of *Escherichia coli, Prevotella* spp., *Clostridium* spp., and *Bacteroides fragilis* [51]. Together, these studies provide valuable insights into the complex interplay between the microbiome and the development of GC, suggesting that microbial dysbiosis and functional changes may play a role in promoting gastric carcinogenesis [52].

The **intratumoral microbiome** of gastric cancer tissue has also been shown to be enriched with oral bacteria, such as *Streptococcus* and *Fusobacterium*, whereas adjacent non-cancerous tissue has higher levels of bacteria that produce lactic acid, such as *Lactococcus lactis* and *Lactobacillus brevis* [53], along with oral species such as *Fusobacterium nucleatum, Veillonella, Leptotrichia, Haemophilus,* and *Campylobacter* [54,55,56]. A microbial dysbiosis index was calculated based on 6 enriched and 18 depleted bacterial genera in GC, and successfully discriminated between GC and NAG patient samples, an AUC of 87% with sensitivity rate of 97%, and provided a false-positive rate of 7.7%. Additionally, bacterial biomarkers such as *P. gingivalis* and *Streptococcus anginosus* have been linked to increased risk of GC and worse clinical outcomes, while *Lactobacillus* has been associated with better prognosis and longer survival [57]. This emerging body of work highlights the potential contribution of microbiome-based biomarker assays that could aid in the diagnosis and treatment planning of upper digestive tract malignancies [58,59].

## 3. Mechanisms of Microbiome Impact on Esophageal and Gastric Carcinogenesis

Although this extensive catalog of epidemiologic evidence linking the microbiome with cancer risk or prognosis is an important first step, it will become vital to understand the underlying biologic mechanisms if this knowledge is to be utilized to improve prevention or treatment of EC/GC. A variety of hypothetical pathways could account for these findings, including direct genotoxicity of microbial products, metabolic alterations stimulated by microbiota, chronic inflammation of upper gastrointestinal epithelial tissues, promotion of epithelial-to-mesenchymal transformation (EMT) by cancer cells, and induction of a tolerogenic phenotype in the tumor immune microenvironment, among others (Figure 2). We begin with gastric cancer, given the vast literature on *H. pylori* as the canonical bacterial carcinogen [60].

Mechanistically, *H. pylori*-related changes in the tumor microenvironment contribute to tumor progression and immune tolerance by causing chronic inflammation [61], genetic alteration including DNA methylation and chromosmomal instability [62,63,64], and immune suppression [65]. *H. pylori*-induced inflammation can create conditions ripe for EMT, by upregulating the expression of pro-angiogenic factors such as vascular endothelial growth factor (VEGF), basic fibroblast growth factor (bFGF), and matrix metalloproteinases (MMPs) [66,67,68]. In addition, *H. pylori* infection can regulate immune checkpoints and modify the tumor microenvironment, which plays a critical role in tumor progression, angiogenesis, metastasis, and drug resistance [69]. The *H. pylori* virulence factor CagA is well established as contributory to gastric carcinogenesis. The CRPIA motif in nonphosphorylated CagA interacts with activated Met, leading to the activation of β-catenin and NF-κB signaling, which promote cell proliferation and inflammation, respectively [70]. Additionally, the Hippo pathway’s effector, YAP1, translocates into the nucleus and cooperates with TEAD to activate the transcription of the inflammatory cytokine IL-1β in *H. pylori*-infected gastric cells, potentially leading to gastric cancer development [71]. Moreover, the attachment of *H. pylori* to gastric epithelial cells leads to an inflammatory immune response that triggers gene alterations in both the adaptive and innate immune systems. These gene alterations involve various components, including interleukins (IL-1β, IL-8), transcription factors (CDX2, RUNX3, TLR1), and DNA repair enzymes [72]. Ultimately, the depletion of specialized glandular tissue and reduction in acid secretion within the gastric tissue leads to the disappearance of *H. pylori* and an increase in the presence of commensal intestinal bacteria such as *Lactobacillus, Enterococci, Carnobacterium, Parvimonas, Citrobacter, Clostridium*, *Achromobacter,* and *Rhodococcus* [48,73,74], creating the potential for microbiome-based biomarkers for changes in the gastric microenvironment.

Whereas *H. pylori* is associated with specific and direct tumorigenic pathways driven by known virulence factors, other less concrete pathways also likely link microbes to GC. GC risk is heavily influenced by diet, with certain types of foods such as broiled and charbroiled animal meats, salt-preserved and smoked foods, and high-fat diets positively associated with GC development due to their ability to produce carcinogens like polycyclic aromatic hydrocarbons and N-nitroso compounds that interact with gastric epithelial cells. The effect of dietary patterns may be mediated by microbial dysbiosis in the stomach to influence GC risk, for example, by producing small molecular metabolites that may induce carcinogenesis. High-fat diets have been shown to promote dysbiosis and the development of intestinal metaplasia (IM) in the stomach, and changes in the gastric microbiota community can affect pathogenesis in the gastric mucosa [75,76,77]. In one expermient, the stomach’s microbial balance was severely disrupted after one week of consuming a high-fat diet (HFD). This change in microbiota was accompanied by an increase in gastric leptin, which led to the development of intestinal metaplasia. When the gastric microbiota from HFD-fed mice was transplanted into recipient mice, they also developed intestinal metaplasia, but with limited effects on the pathogenesis. Interestingly, the abundance of microbial populations was not decreased in HFD-fed db/db mice. Additionally, T3 b-Lepr cKO mice did not develop spontaneous obesity, and their gastric microbiota remained abundant despite being fed an HFD, which resulted in the suppression of intestinal metaplasia. These findings led the authors to propose a model in which gastric leptin signaling plays a role in regulating the gastric microbiota community and influencing pathogenesis in the gastric mucosa [78]. Relatedly, lactic acid bacteria (LAB) are abundant in GC and can produce lactate, which provides malignant cells with a fuel source and makes them more resistant to chemotherapy. Lactic acid bacteria can stimulate the generation of ROS and NOCs, which can cause DNA damage and inhibit apoptosis, playing a role in the development of gastric cancer. Lactic acid-producing bacteria are abundant in patients with GC, and lactate, metabolized by these bacteria, is a source of energy for cancer cells and plays a regulatory role in various aspects of carcinogenesis [79,80,81,82]. Furthermore, lactate inhibits T and NK cells, contributing to tumor immune escape. Thus, despite the common use of *Lactobacillus* species as probiotics, their presence in GC could theorectically be detrimental due to the production of lactate [83,84,85].

In esophageal cancer, recent research using mouse models to investigate *P. gingivalis* as an etiologic agent in ESCC, given the increased abundance of *P. gingivalis* in oral biofilms in patients with this tumor type. *P. gingivalis* was found to be correlated with higher esophageal cancer incidence in a mouse model induced by 4-nitroquinoline 1-oxide and with an increased growth rate of xenograft tumors. There are multiple mechanisms through which *P. gingivalis* can promote tumorigenesis. Firstly, it activates the JAK2 and GSK3β pathways, leading to increased production of the pro-inflammatory cytokine IL-6, which is known to promote EMT and the recruitment of myeloid-derived suppressor cells [86,87,88]. Inhibiting IL-6 signaling in this model attenuated the tumor-promoting effects of *P. gingivalis* in both 4-nitroquinoline 1-oxide-treated mice and xenograft mouse models [89,90]. Secondly, *P. gingivalis* secretes a nucleoside diphosphate kinase (NDK) that inhibits IL-1β production, potentially promoting immune evasion of tumor cells. Additionally, NDK-mediated degradation of ATP suppresses apoptosis dependent on ATP activation of P2X7 receptors, contributing to tumorigenesis [91,92]. Thirdly, *P. gingivalis* inhibits apoptosis in epithelial cells by various mechanisms, including upregulating anti-apoptotic proteins and downregulating pro-apoptotic proteins, manipulating cyclin/CDK activity and reducing the level of the p53 tumor suppressor [93,94]. *P. gingivalis* can promote cellular migration in oral squamous-cell carcinoma (OSCC) cells through the activation of several signaling pathways that induce pro-matrix metalloproteinase (MMP)-9 expression. Additionally, *P. gingivalis* can convert ethanol into the carcinogenic derivative acetaldehyde, which could contribute to the development of some cancers [95,96].

Other studies have identified mechanisms linking *Campylobacter* species to carcinogenesis in EAC. These bacteria have several virulence mechanisms, including toxin production, cellular invasion, and intracellular survival, which suggest that they may play a role in chronic esophageal inflammation that leads to cancer. *Campylobacter concisus*, as an example, has been linked to an increase in the expression of IL-8 and IL-18, both of which are potentially contributory to chronic esophageal inflammation leading to adenocarcinoma [97,98]. Research has revealed that infection with *C. concisus* can affect the regulation of genes and proteins involved in the development of EAC. Specifically, infection with *C. concisus* can lead to downregulation of TGF-β1 signaling, which is important for the development of Barrett’s metaplasia. Conversely, infection with *C. concisus* can upregulate the NF-kβ and STAT3 signaling pathways, which are associated with the inflammation that occurs during the EAC cascade. The SHH-BMP4 signaling axis, which is involved in BE, is also upregulated upon *C. concisus* infection. Interestingly, *C. concisus* infection can upregulate the SOX5 and TWIST1 genes, which contribute to EMT [31,99]. This potential similarity to the role of *H. pylori* in gastric cancer highlights the emerging recognition of *Campylobacter* as a human pathogen [100].

In summary, dysbiosis-associated alterations in the microbiome can influence cancer hallmarks, including inflammation, metabolism, and immune modulation. Understanding these complex interactions offers opportunities for the development of microbiome-based diagnostics and therapeutics. By restoring a healthy microbial balance, it is possible to potentially improve treatment outcomes and enhance the overall management of gastroesophageal cancer. However, further research is needed to unravel the intricate mechanisms underlying microbiome–cancer interactions and to translate these findings into clinical practice.

## 4. Mechanism of Microbiota’s Impact on Host Immune Response

In this section, we review proposed mechanisms whereby the microbiome could exert influence over the tumor immune microenvironment of EC and GC, and thus impact the efficacy of immunotherapy. The microbiome has the ability to promote the activation of antitumor immune responses by influencing antigen presentation and immune cell activation. Microbial molecules, such as pathogen-associated molecular patterns (PAMPs), can interact with pattern-recognition receptors (PRRs) to stimulate the production of proinflammatory cytokines via multiple effector pathways, including the formation of inflammasomes and the activation of nuclear factor-κB (NF-κB), stress kinases, interferon regulatory factors (IRFs), inflammatory caspases, and autophagy. Genotoxins released by certain bacteria, such as reactive oxygen species (ROS), reactive nitrogen species (RNS), and hydrogen sulfide (H2S), can have direct genotoxic effects, while bacterial metabolic actions, including the activation of toxins like acetaldehydes and nitrosamines, can also promote carcinogenesis [101]. In an adaptive response, the local immune reaction to bacterial signals could lead to the activation of dendritic cells and tumor-specific T cells within the tumor microenvironment. Indeed, individual bacterial species have been identified as inducers of anti-tumor immune responses by modulating immune cell function, including the promotion of cytotoxic T cells and natural killer (NK) cells [102,103]. 

The presence of gastric bacteria can interfere with the host’s immune response to GC cells, promoting immune evasion. B cells are found at lower levels in GC than in gastritis, indicating an immune response suppression mechanism that may be facilitated by the microbiome [104,105]. *Fusobacterium*, which is commonly found in gastric cancer, has been found to be positively correlated with the presence of tumor-infiltrating lymphocytes, but it is also known to inhibit T-cell and natural killer cell cytotoxicity, leading to immune evasion and weakening the antitumor effect of tumor-infiltrating lymphocytes [106]. Intratumoral *Methylobacterium*, in particular, has been linked to unfavorable outcomes in gastric cancer patients and has an inverse correlation with the frequency of CD8+ tissue-resident memory T cells in the tumor microenvironment. This bacterium can reduce TGF-β expression and CD8+ TRM cells within the tumor, indicating a potential role in gastric cancer immune evasion [107].

More broadly, dysbiosis and alterations in the microbiome composition can contribute to the development of a maladaptive, immunosuppressive tumor microenvironment. Microbial metabolites, such as short-chain fatty acids (SCFAs) or lipopolysaccharides (LPS), can modulate the differentiation and function of regulatory T cells (T_reg_) and myeloid-derived suppressor cells (MDSCs), thereby promoting immune tolerance. Additionally, the microbiome can influence the expression of immune checkpoints within the tumor microenvironment [108], which could directly influence the potential therapeutic efficacy in a given patient of immune checkpoint inhibitors (ICIs) such as anti-PD-1/PD-L1 or anti-CTLA-4 antibodies. Specific bacterial taxa have been associated with increased response rates or improved survival in patients receiving ICIs. Furthermore, the microbiome may also influence other immunotherapeutic approaches such as adoptive T cell therapy, cancer vaccines, or chimeric antigen receptor (CAR) T cell therapy [109]. The microbiome can impact the priming and expansion of adoptively transferred T cells, as well as the systemic immune responses necessary for effective immunotherapy [110].

The exploration of microbial genes and their potential interactions with immunosuppressant therapies marks a pivotal advancement in comprehending the intricate interplay between the human microbiome and therapeutic interventions [111]. An illustrative example lies in the realm of drug metabolism, where bacterial enzymes encoded by microbial genes can profoundly impact immunosuppressant bioavailability and effectiveness. Notably, the gut bacterium Eggerthella lenta harbors genes encoding β-glucuronidases that can deconjugate glucuronide-bound drugs, thus reversing their inactivation and potentially altering therapeutic levels [112]. This enzymatic activity exemplifies how microbial genetic elements can directly influence drug pharmacokinetics, underscoring the need to consider such interactions in therapeutic regimens.

Moreover, specific microbial genetic elements, like those found in Bacteroides fragilis, have been linked to the synthesis of metabolites with immunomodulatory properties. The polysaccharide A (PSA) produced by this bacterium has demonstrated the ability to regulate immune responses by promoting anti-inflammatory cytokine production [113]. Consequently, when subjected to immunosuppressant therapies, the presence or absence of these microbial genes can potentially impact the balance between immune suppression and the maintenance of protective immune responses.

The forthcoming study is dedicated to conducting an exhaustive survey of literature, aimed at identifying and understanding microbial genes associated with immunosuppressant interactions. By synergizing this knowledge with our broader research framework, we aspire to unveil the intricate molecular mechanisms through which microbial genetic elements orchestrate immunosuppressant responses. This endeavor, empowered by specific bacterial gene examples, not only advances our comprehension of the multifaceted relationships between the microbiome, microbial genetics, and treatment outcomes but also lays the groundwork for tailoring therapeutic approaches that harness microbial influences to optimize treatment precision. In embracing this innovative approach, we anticipate contributing substantive insights to the evolving paradigms of immunotherapy and personalized medicine [114].

## 5. Current Approaches to Treating Advanced Gastroesophageal Cancers and Clinical Trials Involving Checkpoint Inhibitors

Before immunotherapy became a viable option, patients with advanced or metastatic EC and GC were mainly treated with palliative cytotoxic chemotherapy and radiation, but these had poor response rates, limited survival benefits, and few predictive biomarkers to guide treatment [115,116]. However, the introduction of immune checkpoint inhibition (ICI) therapy has transformed the treatment of gastrointestinal tumors, building on the early successes of melanoma treatment. The treatment landscape for gastroesophageal cancers continues to evolve rapidly. Despite this, only a minority of patients meet eligibility criteria for ICI therapy with the current FDA-approved options for various malignancies, which include PD-1 blockers (nivolumab, pembrolizumab, and cemiplimab), PD-L1 blockers (atezolizumab, avelumab, and durvalumab), and CTLA-4 blockers (ipilimumab) [7].

Most patients with advanced gastroesophageal cancers will receive first-line immunotherapy, while some still receive chemotherapy. Untreated advanced esophageal cancer is responsive to immunomodulation with pembrolizumab or Nivolumab + chemotherapy, and the combination of chemotherapy and immunotherapy improves survival rates compared with chemotherapy alone based on KEYNOTE-590, CheckMate-648, and CheckMate-649 [117]. Two second-line immunotherapies are approved by the FDA [118,119,120]. Nivolumab, a monoclonal anti-PD-1 antibody, has demonstrated improved overall survival as a later-line therapy in unselected GC patients in the ATTRACTION-3 study, regardless of the tumor PD-1 expression [121]. In addition, it has shown promise in combination with chemotherapy as a first-line therapy in the global CheckMate-649 study. Meanwhile, pembrolizumab, another anti-PD-1 antibody, has been effective as a single agent in tumors with high microsatellite instability or high tumor mutational burden [122]. More recently, in the KEYNOTE-811 study, the combination of pembrolizumab with trastuzumab and chemotherapy for HER2-positive GC/GEJ cancers demonstrated a significant improvement in response rate [123]. Currently, two immunotherapies have histology agnostic indications regardless of tumor location: Pembrolizumab and Dostarlimab. Dostarlimab, a humanized anti-PD-1 monocolonal antibody, demonstrated a favorable overall response rate in the GARNET trial, in a small subtype of patients with advanced GC harboring mismatch repair deficiency (d-MMR) [124]. Hence, these three anti-PD-1 ICI agents have been approved for use in the respective settings in patients with GC in the United States.

However, ICI monotherapy is only sometimes effective, and high primary and secondary resistance rates have led to combination strategies with other types of therapy to improve response rates [121]. Ongoing trials are investigating the combination of ICIs with chemotherapy, anti-angiogenesis agents, anti-HER2 agents, and radiotherapy. In a phase II trial (NCT03878472), a combination of ICI, anti-angiogenic, and cytotoxic therapy was tested for neoadjuvant/conversion treatment of cT3-4bN+M0 GC. The study found significant pathological responses, potential biomarkers, and limited toxicity and complications. Additionally, pembrolizumab, in combination with Lenvatinib (an oral multikinase inhibitor), showed promise in treating advanced GC in a phase II Japanese study [123]. These results suggest the need for further validation in large randomized trials [125]. Combining immune checkpoint inhibitors with anti-HER-2 drugs is a promising new treatment concept for HER-2+ GEC with good efficacy and tolerable side effects, according to clinical trials [126].

Predictive biomarkers can help identify patients who may benefit from targeted treatments like PD-1/PD-L1 inhibitors, since not all patients respond to ICI therapy [127]. Relevant biomarkers for selection of EC and GC patients eligible for ICI include PD-L1 immunohistochemistry, microsatellite instability (MSI), mismatch repair deficiency (MMR), tumor mutational burden (TMB), Epstein–Barr Virus (EBV) positivity, immune gene signatures, overexpression of HER2, and epigenetic alterations [128]. Predictive biomarkers are also being studied to identify patients at higher risk of developing drug resistance and improve treatment decisions [129]. A systematic review and meta-analysis of 17 phase III randomized clinical trials involving 11,166 patients with gastroesophageal cancer revealed that the most significant predictors of improved overall survival benefit from ICI were microsatellite instability and tissue-based PD-L1 expression (tumor proportion score/TPS for squamous and combined positive score/CPS for adenocarcinoma) [130,131]. Given the potential toxicity of ICI therapy, further development of response biomarkers is warranted in order to improve patient selection (Table 1).

## 6. Connection between Microbiota, the Immune System Response of the Host, and the Impact on Cancer Immunotherapy and Treatment Toxicity

Considering therapeutic strategies targeting the microbiome, various approaches can be employed to modulate the microbiome composition. These include the use of probiotics, prebiotics, antibiotics, fecal microbiota transplantation (FMT), and administration of microbial metabolites. These interventions aim to restore microbial balance, enhance antitumor immune responses, and improve the efficacy of immunotherapy in EC and GC. Since the microbiome has an influence on the tumor immune microenvironment, manipulation of the microbiome has emerged as a potential target for enhancing the response to ICIs [132]. In mice, the composition of the gut microbiome has been shown to affect the host response to PD-1/PD-L1 blockade or CTLA-4 inhibition. *Bifidobacteria*, for example, have been found to be enhanced in mice with slow tumor growth and better responses to anti-PD-1 therapy. Groundbreaking work in clinical studies has confirmed that the gut microbiome composition may impact the effectiveness of immune checkpoint therapy in a number of tumor types, including sites quite distant from the gastrointestinal tract [133]. Prospective studies are currently being conducted to explore dietary modulation, probiotics, antibiotics, and fecal microbiota transplantation as potential methods for manipulating the gut microbiome and promoting a more favorable immune response [134]. FMT, specifically, could be leveraged to enhance the systemic and antitumor immune response in cancer patients receiving ICIs [135,136,137,138,139].

Although promising, FMT is plagued with a number of logistical challenges, not the least of which is that it contains a mixture of microbial species and other elements, only some of which are likely to account for the desired effects. In order to achieve precise and consistent results, there is a need to define the specific active constituents of FMT. Studies have demonstrated that the administration of discrete species of live bacteria, such as *Bifidobacteria, Lactobacillus, Propionibacterium*, and *Streptococcus thermophilus*, in combination with ICIs, can significantly improve the outcomes of cancer patients who receive immunotherapy, by enhancing the efficacy of both CTLA4 and anti-PD-L1 immunotherapy for four different types of cancer [140,141,142]. The success of anti-CTLA-4 therapy has been linked to the presence of certain gut microbiota populations such as *Bacteroides thetaiotaomicron, Burkholderiales*, and *Bacteroides fragilis*. The reintroduction of *B. fragilis* cells and/or polysaccharides, or the transfer of *B. fragilis*-specific T cells, could enhance the success of the therapy and decrease immune-mediated colitis. This was achieved through the activation of T_h_1 cells that cross-react with bacterial antigens and tumor neoantigens [143]. Another strategy to promote outgrowth of beneficial microbes involves prebiotics such as resistant starch, oligofructose, and inulin. These elements support the growth of organisms that produce metabolites, such as short-chain fatty acids (SCFAs) and inosine, that are known to enhance tumor-cell killing efficacy by promoting both effector T lymphocyte subsets in the context of a favorable cytokine milieu [144,145,146]. This strategy, if shown to be effective in combination regimens with other immunotherapy modalities, could be particularly appealing since it avoids the logistic complexities of FMT or live bacterial therapies.

*H. pylori*’s virulence factors can also function as immunogens or adjuvants to induce or enhance immune responses, suggesting their potential use in vaccine development [147]. Research conducted by Shi et al. has revealed two important aspects of the possible interactions between *H. pylori* and cancer immunotherapies. Firstly, purified or cloned elements of *H. pylori*, such as HP-NAP, CagA, VacA, BabA, and HspA, can function as potent adjuvants, effectively enhancing tumor responses. Secondly, *H. pylori* infection may affect the efficacy of antitumor immunity triggered by ICIs by modulating host immune responses [148]. Che et al. showed an association between *H. pylori* infection and the outcome of immunotherapy for GC patients, where patients in the *H. pylori*-positive group were at a higher risk of nonclinical response to an anti-PD-1 antibody compared to the *H. pylori*-negative group [149]. It remains to be seen whether vaccine strategies incorporating *H. pylori* or other microbial elements could gain traction in active treatment regimens for upper GI malignancies.

The microbiome holds great promise as a potential trove of biomarkers for ICI response. For example, Sunakawa et al. of the Japan Clinical Cancer Research Organization (JACCRO) studied the effect of gut microbiome gene expression as a predictor for ICI efficacy in 501 advanced GC treated with Nivolumab monotherapy. The authors reported that upregulation of the bacterial invasion of the epithelial cell pathway (KEGG) was related to disease progression. Furthermore, an exploratory analysis indicated that bacteria of the genus *Odoribacter* and *Veillonella* were associated with tumor response to Nivolumab [150]. In addition to influencing efficacy of immunotherapy, the microbiome may also modulate or predict toxicity. A study of 95 patients with advanced gastrointestinal cancers who underwent immunotherapy analyzed the gut microbiome’s association with immune-related adverse events (irAEs) using metagenome sequencing of baseline fecal samples and identified bacterial species and metabolic pathways that could be linked to the development of irAEs in patients with gastric, esophageal, and colon cancers. *Ruminococcus callidus* and *Bacteroides xylanisolvens* were more prevalent in patients who did not experience severe irAEs. Various microbial metabolic pathways involved in the urea cycle, including citrulline and arginine biosynthesis, were linked to irAEs. Furthermore, distinct gut microbiota profiles were associated with irAEs in different cancer types, as well as toxicity in specific organs and the endocrine system [151]. 

The advantages of modulating the microbiome composition over other immunomodulators, such as the TIGIT antibody, are profound and multifaceted. Microbiome modulation offers a distinctive set of benefits that stem from its holistic and systemic impact on both the tumor microenvironment (TME) and the intricate web of immune responses [152]. In contrast to traditional immunomodulatory agents like monoclonal antibodies or pharmaceutical compounds, which often target specific molecules or pathways, microbiome modulation exerts its effects across a spectrum of interactions [153]. By addressing the underlying factors contributing to TME dysregulation, microbiome-based interventions could potentially engender sustainable and enduring enhancements in immune function. The broad-ranging influence of microbiome modulation extends beyond singular molecular targets, aiming to recalibrate the complex interplay between host physiology and microbial communities. It is worth acknowledging that each approach, whether microbiome modulation or traditional immunomodulation, boasts its own merits and confronts unique challenges. Nevertheless, the convergence of these strategies presents a compelling avenue for synergistic outcomes that transcend the limitations of individual interventions.

## 7. Summary and Recommendations for Improving the Effectiveness of Microbiota in Cancer Immunotherapy

While few of the findings reviewed here are sufficiently mature to aid in the development of broadly applicable interventions at the present time, emerging knowledge regarding the impact of tumor and host microbiome on tumor biology and treatment response will provide exciting avenues to improve the care of patients with gastric and esophageal cancer. Overall, these interventions aim to alter the composition of the gut microbiome to promote a more favorable immune response, potentially reducing treatment resistance and improving outcomes for patients receiving cancer immunotherapy. However, more research is needed to fully understand the potential of these interventions, including their safety and efficacy, and to identify which patients may benefit the most from such interventions [146]. While most research has focused on the intestinal microbiota, little attention has been paid to microbiomes at other barrier sites, such as the skin and oral microbiota, which may also play a critical role. There are challenges in identifying reliable biomarkers and developing combination therapies that can enhance the efficacy of immunotherapy for gastroesophageal cancer. The composition of microbiota has an impact on tumor development, and screening for microbiota composition could potentially identify people at a higher risk of developing cancer. Modifying the microbiota of these individuals could also have implications for cancer prevention and treatment, especially with immunotherapy. The potential role of microorganisms other than bacteria, such as fungi, viruses, and protozoa, is also beginning to be explored.

## 8. Conclusions

In conclusion, this article highlights the increasingly recognized influence of the microbiome on gastroesophageal cancer biology, specifically focusing on its associations with alterations in various microbiota and the tumor microenvironment. As immunotherapy gains traction as a treatment option for gastroesophageal cancer, understanding the microbiome’s impact on the immunosuppressive tumor microenvironment is crucial. Dysbiosis and microbial diversity shifts have been linked to treatment resistance, while specific microbial profiles correlate with improved outcomes. The potential for microbiome-based biomarkers and interventions, such as microbiota-targeted therapies and dietary modifications, offers a pathway to enhance immunotherapy efficacy by reshaping the tumor microenvironment. Continued research and rigorous clinical trials are needed to provide mechanistic insights and validate the efficacy of microbiome-focused interventions, holding promise for personalized approaches to improve gastroesophageal cancer treatment outcomes. 

## Figures and Tables

**Figure 1 cancers-15-04426-f001:**
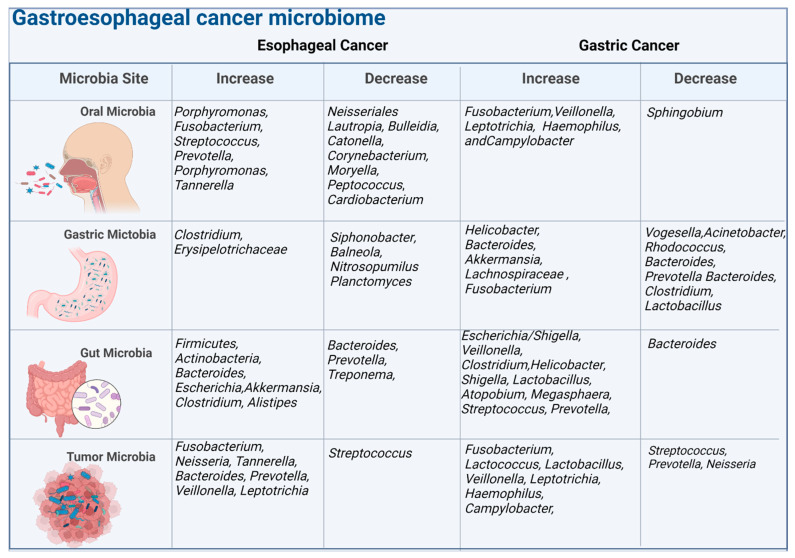
The figure illustrates the microbiome composition at different body sites, including the oral cavity, gastric region, gut, and tumor. It highlights the associations between the microbiome and esophageal cancer and gastric cancer, indicating increases or decreases in specific bacteria. The analysis is conducted at the genus level, representing groups of related bacteria.

**Figure 2 cancers-15-04426-f002:**
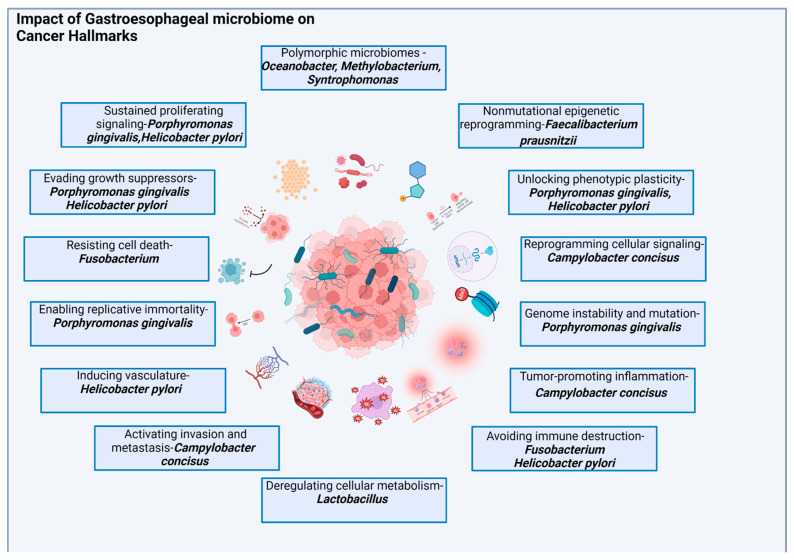
The figure illustrates the impact of the microbiome on cancer hallmarks and tumor biology. It highlights the specific effects of bacteria on various aspects of cancer progression, including sustained proliferating signaling, evading growth suppressors, resisting cell death, enabling replicative immortality, inducing vasculature, activating invasion and metastasis, deregulating cellular metabolism, avoiding immune destruction, promoting tumor-promoting inflammation, genome instability and mutation, reprogramming cellular signaling, unlocking phenotypic plasticity, non-mutational epigenetic reprogramming, and polymorphic microbiomes. Detailed explanations of these effects are provided in the body of the article.

**Table 1 cancers-15-04426-t001:** Checkpoint Inhibitor Approaches and Clinical Trials: Summary of Strategies, Trials, and Results. Legend: GEJ—Gastroesophageal, SCC—Squamous-cell carcinoma, AC—Adenocarcinoma, Chemo—Chemotherapy, 5FU—5-Fluorouracil, Cis—Cisplatin, Pembro—Pembrolizumab, Nivo—Nivolumab, Ipi—Ipilimumab, OS—Overall survival, PFS—Progression-free survival, Her2—Human epidermal growth factor receptor 2.

Name of the Trial	Disease Site	Trial Phase	Arms of the Study	Primary Endpoint	Results
KEYNOTE-590	Advanced Esophageal or GEJ SCC	Phase III	Chemo + Pembro vs. Chemo + Placebo	OS, PFS	OS: 13.9 vs. 8.8 months
CHECKMATE-648	Advanced Esophageal or GEJ SCC	Phase III	Chemo + Nivo vs. Ipi + Nivo vs. Chemo alone	OS, PFS	OS: 13.2 vs. 12.7 vs. 10.7 months
CHECKMATE-649	Advanced Esophageal or GEJ AC	Phase III	Chemo + Nivo vs. Ipi + Nivo vs. Chemo alone	OS, PFS	OS: 13.1 vs. 11.1 months
ATTRACTION-3	Advanced Esophageal or GEJ SCC	Phase III	Nivo vs. Chemo	OS	OS: 10.9 vs. 8.4 months
KEYNOTE-811	Advanced gastric or GEJ AC, Her2 positive	Phase III	(Chemo + Trastuzumab) + Pembro vs. Placebo	OS, PFS	Study ongoing

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
