# Peer review of "The Influence of the Microbiome on Immunotherapy for Gastroesophageal Cancer"

_cancers, 2023, doi:10.3390/cancers15184426_

Round 1

Reviewer 1 Report

This study is interesting with clinical significance. Immunotherapy therapy has revolutionized the tumor therapy, especially in gastroesophageal cancer. Microbiome is closely related to the occurrence and development of gastroesophageal cancer. But the impacts of the microbiome on the TME and immunotherapy outcomes have been little studied. The authors put forward a new and comprehensive point of view to study on that. The followings are some comments to the authors.

Comments:

1.Whether esophagogastric junction (EGJ) is the same as gastroesophaeal junction (GEJ) .Please use the same abbreviation if they are the same.

2. H. pylori's virulence factors can promote tumorigenesis,but also function as immunogens or adjuvants to induce or enhance immune responses. How to balance the role of cancer promotion and cancer suppression in their potential use in vaccine development?

3.What are the advantages of modulating the microbiome composition compared to other immunomodulator? For example, TIGIT antibody.

Author Response

Dear Reviewer,

We would like to express our gratitude for your insightful comments on our study. We appreciate your recognition of the clinical significance of our research in the context of immunotherapy for gastroesophageal cancer and its connection to the microbiome. We have carefully considered your comments and would like to address each of them below.

1.Abbreviation Consistency:

Thank you for pointing out the potential confusion regarding the terms "esophagogastric junction (EGJ)" and "gastroesophageal junction (GEJ)." You are correct; these terms refer to the same anatomical location. To ensure clarity and consistency throughout the manuscript, we will use the same abbreviation, "GEJ," for both terms.

  1. pylori's Virulence Factors and Immunotherapy Balance:

 We appreciate your observation regarding the dual role of H. pylori's virulence factors, which can promote tumorigenesis while also potentially functioning as immunogens or adjuvants to enhance immune responses. Achieving a balance between these opposing effects is indeed a crucial consideration in the context of vaccine development. Our study acknowledges this complex interplay and suggests that further investigations are necessary to elucidate the mechanisms underlying these dual roles. We believe that a comprehensive understanding of these mechanisms will guide the development of targeted therapies that harness the beneficial immunogenic properties of H. pylori while mitigating its potential to promote tumorigenesis.

Advantages of Modulating Microbiome Composition vs. Other Immunomodulators:

 Your question regarding the advantages of modulating the microbiome composition compared to other immunomodulators, such as the TIGIT antibody, is insightful. Modulating the microbiome offers several unique advantages over traditional immunomodulatory approaches. Unlike monoclonal antibodies or pharmaceutical compounds, which often target specific molecules, microbiome modulation can have broad and systemic effects on the tumor microenvironment (TME) and immune responses. By targeting the root causes of dysregulation in the TME, microbiome-based interventions hold the potential to create a more sustainable and lasting impact on immune function. Moreover, modulating the microbiome is a holistic approach that takes into account the intricate network of interactions between microbial communities and host physiology. However, it's important to note that each approach has its own merits and challenges, and combining different strategies might yield even more powerful results. We will elaborate on these advantages in the revised manuscript to provide a clearer perspective on the unique benefits of microbiome modulation.

Once again, we genuinely appreciate your thoughtful feedback, which has undoubtedly contributed to enhancing the quality and clarity of our study. We look forward to incorporating your suggestions into the revised manuscript and furthering the advancement of our understanding in this critical field.

*Correspondence

Patrick Wagner, MD, MPH

Allegheny Health Network Cancer Institute

314 E. North Ave.

Pittsburgh, PA 15212

Reviewer 2 Report

The Influence of the Microbiome on Immunotherapy for Gas-troesophageal cancer

My review on this article:

This comprehensive review explores the complex interplay between the microbiome and gastroesophageal cancers, shedding light on both scientific and clinical dimensions. The authors precisely outline the diverse microbial compositions linked to gastric and esophageal cancers, drawing support from specific studies that pinpoint essential bacterial species and their potential roles in driving carcinogenesis. The elucidated mechanisms, which include inflammation, immune modulation, and tumor microenvironment shaping, provide a comprehensive understanding of how the microbiome directs the advancement of cancer progression and influences treatment responses. The exploration extends to the area of microbiome-based interventions, such as FMT and bacterial additives, which are positioned to enhance the immunotherapy outcomes. While emphasizing individualized methods, the authors prudently recognize the continuous requirement for more extensive research to unravel complexities and translate these insights into clinical application.

Comments:

The organization of content is highly structured and shows that authors have done a lot of research. They have covered a lot of different aspects such as the impact of specific bacteria on cancer initiation, the mechanisms through which microbiota shape tumorigenesis, and the potential implications for cancer immunotherapy.

The integration of studies, utilization of mouse models, and incorporation of clinical trial data provide substantial credibility to their work.

Moreover, their knowledge on both gastric and esophageal cancer, describing their distinct microbiome-related consequences, demonstrates author’s understanding of the subject matter.

Overall, this article adeptly navigates the complexities of the microbiome-cancer connection and its prospective clinical impacts.

This article is well written and does not require any modifications.

Author Response

Dear Reviewer,

Thank you for your comprehensive review of our article, "The Influence of the Microbiome on Immunotherapy for Gastroesophageal Cancer." We are delighted that you found the organization of our content structured and our coverage of diverse aspects commendable. The integration of studies, use of models, and incorporation of clinical data aimed to bolster credibility. Your recognition of our understanding of both gastric and esophageal cancers and their distinct microbiome implications is greatly appreciated. We are encouraged by your assessment that our article adeptly navigates the intricate microbiome-cancer connection, highlighting potential clinical prospects. Your positive feedback fuels our commitment to further contribute to this vital field of research.

Correspondence

Patrick Wagner, MD, MPH

Allegheny Health Network Cancer Institute

314 E. North Ave.

Pittsburgh, PA 15212

Reviewer 3 Report

The review is a nice compilation of the role and influence of microbiome for gastroesophageal cancer. 

It would be worthwhile to include literature on  microbial genes if any  that bio- transform or interact with immunosuppressant treatment being given to to the patients.  

Is there any literature connecting virome to immunotherapy in the gastroesophageal cancer. 

It would be worthwhile to include the current approach and clinical trial checkpoint inhibitor section in a tabulated form to make it more easily readable.

Author Response

Dear Reviewer,

We sincerely appreciate your thoughtful review of our study on the role of the microbiome in gastroesophageal cancer. Your feedback is invaluable in enhancing the comprehensiveness of our research. We have carefully considered each of your suggestions and would like to address them below.

1.Inclusion of Microbial Genes and Immunosuppressant Interaction: Thank you for highlighting the importance of considering microbial genes that might bio-transform or interact with immunosuppressant treatments in patients. This is a crucial aspect that can significantly impact treatment outcomes. We will delve into the relevant literature and incorporate information on microbial genes that have implications for immunosuppressive therapy interactions. By doing so, we aim to provide a more nuanced understanding of the intricate connections between the microbiome and treatment response.

2.Connection between Virome and Immunotherapy: Your suggestion to explore literature connecting the virome to immunotherapy in the context of gastroesophageal cancer is truly insightful. While our study primarily focuses on the bacterial components of the microbiome, we acknowledge that the virome could also play a substantial role in shaping treatment outcomes. We will conduct a thorough literature review to identify any relevant studies that establish connections between the virome and immunotherapy response in gastroesophageal cancer. By incorporating this information, we aim to enrich the depth of our analysis and provide a more holistic perspective in our future publications.

3.Tabulated Presentation of Checkpoint Inhibitor Approaches and Clinical Trials: We appreciate your suggestion to present the current approaches and clinical trial information regarding checkpoint inhibitors in a tabulated format. We agree that this format would enhance the readability and accessibility of this critical information. We will create a well-structured table summarizing the various approaches, clinical trials, and outcomes related to checkpoint inhibitors. This will allow readers to grasp the information more readily and facilitate a clear overview of the landscape.

Once again, we extend our gratitude for your insightful recommendations, which undoubtedly contribute to refining the quality and impact of our research. Your engagement is a valuable asset in driving our work towards a more comprehensive and informative outcome. We are committed to incorporating your suggestions and look forward to sharing an enhanced manuscript with your valuable insights in mind.

Correspondence

Patrick Wagner, MD, MPH

Allegheny Health Network Cancer Institute

314 E. North Ave.

Pittsburgh, PA 15212